# Bayesian Inference via Sparse Hamiltonian Flows

**Naitong Chen**     **Zuheng Xu**     **Trevor Campbell**
Department of Statistics
University of British Columbia
`[naitong.chen | zuheng.xu | trevor]@stat.ubc.ca`

## Abstract

A Bayesian coreset is a small, weighted subset of data that replaces the full dataset during Bayesian inference, with the goal of reducing computational cost. Although past work has shown empirically that there often *exists* a coreset with low inferential error, efficiently constructing such a coreset remains a challenge. Current methods tend to be slow, require a secondary inference step after coreset construction, and do not provide bounds on the data marginal evidence. In this work, we introduce a new method—*sparse Hamiltonian flows*—that addresses all three of these challenges. The method involves first subsampling the data uniformly, and then optimizing a Hamiltonian flow parametrized by coreset weights and including periodic *momentum quasi-refreshment steps*. Theoretical results show that the method enables an exponential compression of the dataset in a representative model, and that the quasi-refreshment steps reduce the KL divergence to the target. Real and synthetic experiments demonstrate that sparse Hamiltonian flows provide accurate posterior approximations with significantly reduced runtime compared with competing dynamical-system-based inference methods.

## 1   Introduction

Bayesian inference provides a coherent approach to learning from data and uncertainty assessment in a wide variety of complex statistical models. Two standard methodologies for performing Bayesian inference in practice are Markov chain Monte Carlo (MCMC) [1; 2; 3, Ch. 11,12] and variational inference (VI) [4, 5]. MCMC simulates a Markov chain that targets the posterior distribution. In the increasingly common setting of large-scale data, most exact MCMC methods are intractable. This is essentially because simulating each MCMC step requires an (expensive) computation involving each data point, and many steps are required to obtain inferential results of a reasonable quality. To reduce cost, a typical approach is to perform the computation for a random subsample of the data, rather than the full dataset, at each step [6–10] (see [11] for a recent survey). However, recent work shows that the speed benefits are outweighed by the drawbacks; uniformly subsampling at each step causes MCMC to either mix slowly or provide poor inferential approximation quality [11–15]. VI, on the other hand, posits a family of approximations to the posterior and uses optimization to find the closest member, enabling the use of scalable stochastic optimization algorithms [16, 17]. While past work involved simple parametric families, recent work has developed flow families based on Markov chains [18, 19]—and in particular, those based on Langevin and Hamiltonian dynamics [20–25]. However, because these Markov chains are typically designed to target the posterior distribution, each step again requires a computation involving all the data, making KL minimization and sampling slow. Repeated subsampling to reduce cost has the same issues that it does in MCMC.

Although repeated subsampling in each step of a Markov chain (for MCMC or VI) is not generally helpful, recent work on *Bayesian coresets* [26] has provided empirical evidence that there often exists a *fixed* small, weighted subset of the data—a coreset—that one can use to replace the full dataset in a standard MCMC or VI inference method [27]. In order for the Bayesian coreset approach to be

36th Conference on Neural Information Processing Systems (NeurIPS 2022).

practically useful, one must (1) find a suitable coreset that provides a good posterior approximation; and (2) do so quickly enough that the speed-up of inference is worth the time it takes to find the coreset. There is currently no option that satisfies these two desiderata. Importance weighting methods [26] are fast, but do not provide adequate approximations in practice. Sparse linear regression methods [28–30] are fast and sometimes provide high-quality approximations, but are very difficult to tune well. And sparse variational methods [27, 31] find very high quality coreset approximations without undue tuning effort, but are too slow to be practical.

This work introduces three key insights. First, we can uniformly subsample the dataset once to pick the points in the coreset (the weights still need to be optimized). This selection is not only significantly simpler than past algorithms; we show that it enables constructing an *exact* coreset—with KL divergence 0 to the posterior—of size $O(\log_2(N))$ for $N$ data points in a representative model (Proposition 3.1). Second, we can then construct a normalizing flow family based on Hamiltonian dynamics [21, 22, 32] that targets the coreset posterior (parametrized by coreset weights) rather than the expensive full posterior. This method address all of the current challenges with coresets: it enables tractable i.i.d. sampling, provides a known density and normalization constant, and is tuned using straightforward KL minimization with stochastic gradients. It also addresses the inefficiency of Markov-chain-based VI families, as the Markov chain steps are computed using the inexpensive coreset posterior density rather than the full posterior density. The final insight is that past momentum tempering methods [21] do not provide sufficient flexibility for arbitrary approximation to the posterior, even in a simple setting (Proposition 3.2). Thus, we introduce novel periodic *momentum quasi-refreshment* steps that provably reduce the KL objective (Propositions 3.3 and A.2). The paper concludes with real and synthetic experiments, demonstrating that sparse Hamiltonian flows compare favourably to both current coreset compression methods and variational flow-based families. Proofs of all theoretical results may be found in the appendix.

It is worth noting that Hamiltonian flow posterior approximations based on a weighted data subsample were also developed in concurrent work in the context of variational annealed importance sampling [33], and subsampling prior to weight optimization was developed in concurrent work on MCMC [34]. In this work, we focus on incorporating Bayesian coresets into Hamiltonian-based normalizing flows to obtain fast and accurate posterior approximations.

## 2 Background

### 2.1 Bayesian coresets

We are given a target probability density $\pi(\theta)$ for variables $\theta \in \mathbb{R}^d$ that takes the following form:

$$\pi(\theta) = \frac{1}{Z} \exp \left( \sum_{n=1}^{N} f_n(\theta) \right) \pi_0(\theta).$$

In a Bayesian inference problem with i.i.d. data, $\pi_0$ is the prior density, the $f_n$ are the log-likelihood terms for $N$ data points, and the normalization constant is in general not known. The goal is to take samples from the distribution corresponding to density $\pi(\theta)$.

In order to avoid the $\Theta(N)$ cost of evaluating $\log \pi(\theta)$ or $\nabla \log \pi(\theta)$ (at least one of which must be conducted numerous times in most standard inference algorithms), *Bayesian coresets* [26] involve replacing the target with a surrogate density of the form

$$\pi_w(\theta) = \frac{1}{Z(w)} \exp \left( \sum_{n=1}^{N} w_n f_n(\theta) \right) \pi_0(\theta),$$

where $w \in \mathbb{R}^N$, $w \geq 0$ are a set of weights. If $w$ has at most $M \ll N$ nonzeros, the $O(M)$ cost of evaluating $\log \pi_w(\theta)$ or $\nabla \log \pi_w(\theta)$ is a significant improvement upon the original $\Theta(N)$ cost.

The baseline method to construct a coreset is to draw a uniformly random subsample of $M$ data points, and give each a weight of $N/M$; although this method is fast in practice, it typically generates poor posterior approximations. More advanced techniques generally involve significant user tuning effort [26, 28–30]. The current state-of-the-art black box approach formulates the problem as variational inference [27, 31] and provides a stochastic gradient scheme using samples from $\pi_w$,

$$w^\star = \operatorname*{arg\,min}_{w \in \mathbb{R}_+^N} \mathrm{D_{KL}} \left( \pi_w || \pi \right) \quad \text{s.t.} \quad \|w\|_0 \leq M.$$

Empirically, this method tends to produce very high-quality coresets [27]. However, to estimate the gradient at each iteration of the optimization, we require MCMC samples from the weighted coreset posterior at that iteration. While generating MCMC samples from a sparse coreset posterior is not expensive, it is difficult to tune the algorithm to ensure the quality of these MCMC samples across iterations (as the weights, and thus the coreset posterior, change each iteration). The amount of tuning effort required makes the application of this method too slow to be practical. Once the coreset is constructed, all of the aforementioned methods require a secondary inference algorithm to take draws from $\pi_w$. Further, since $Z(w)$ is not known in general, it is not tractable to use these methods to bound the marginal evidence $Z$.

## 2.2 Hamiltonian dynamics

In this section we provide a very brief overview of some important aspects of a special case of Hamiltonian dynamics and its use in statistics; see [35] for a more comprehensive overview. The differential equation below in Eq. (1) describes how a (deterministic) Hamiltonian system with position $\theta_t \in \mathbb{R}^d$, momentum $\rho_t \in \mathbb{R}^d$, differentiable negative potential energy $\log \pi(\theta_t)$, and kinetic energy $\frac{1}{2}\rho_t^T \rho_t$ evolves over time $t \in \mathbb{R}$:

$$\frac{\mathrm{d}\rho_t}{\mathrm{d}t} = \nabla \log \pi(\theta_t) \qquad \frac{\mathrm{d}\theta_t}{\mathrm{d}t} = \rho_t. \tag{1}$$

For $t \in \mathbb{R}$, define the mappings $H_t : \mathbb{R}^{2d} \to \mathbb{R}^{2d}$ that take $(\theta_s, \rho_s) \mapsto (\theta_{s+t}, \rho_{s+t})$ under the dynamics in Eq. (1). These mappings have two key properties that make Hamiltonian dynamics useful in statistics. First, they are invertible, and preserve volume in the sense that $|\det \nabla H_t| = 1$. In other words, they provide tractable density transformations: for any density $q$ on $\mathbb{R}^{2d}$ and pushforward $q_t$ on $\mathbb{R}^{2d}$ under the mapping $H_t$, we have that $q_t(\cdot, \cdot) = q\left(H_t^{-1}(\cdot, \cdot)\right)$. Second, the *augmented target density* $\bar{\pi}(\theta, \rho)$ on $\mathbb{R}^{2d}$ corresponding to independent draws from $\pi$ and $\mathcal{N}(0, I)$,

$$\bar{\pi}(\theta, \rho) \propto \pi(\theta) \cdot \exp\left(-\frac{1}{2}\rho^T \rho\right),$$

is invariant under the mappings $H_t$, i.e., $\bar{\pi}(H_t(\cdot, \cdot)) = \bar{\pi}(\cdot, \cdot)$. Given these properties, Hamiltonian Monte Carlo [35, 36] constructs a Gibbs sampler for $\bar{\pi}$ that interleaves Hamiltonian dynamics with periodic stochastic momentum refreshments $\rho \sim \mathcal{N}(0, I)$. Upon completion, the $\rho$ component of the samples can be dropped to obtain samples from the desired target $\pi$.

In practice, one approximately simulates the dynamics in Eq. (1) using the leapfrog method, which involves interleaving three discrete transformations with step size $\epsilon > 0$,

$$\hat{\rho}_{k+1} = \rho_k + \frac{\epsilon}{2}\nabla \log \pi(\theta_k) \qquad \theta_{k+1} = \theta_k + \epsilon\hat{\rho}_{k+1} \qquad \rho_{k+1} = \hat{\rho}_{k+1} + \frac{\epsilon}{2}\nabla \log \pi(\theta_{k+1}). \tag{2}$$

Denote the map constructed by applying these three steps in sequence $T_\epsilon : \mathbb{R}^{2d} \to \mathbb{R}^{2d}$. As the transformations in Eq. (2) are all shear, $T_\epsilon$ is also volume-preserving, and for small enough step size $\epsilon$ it nearly maintains the target invariance. Note also that evaluating a single application of $T_\epsilon$ is of $O(Nd)$ complexity, which is generally expensive in the large-data (large-$N$) regime.

## 2.3 VI via Hamiltonian dynamics

Since the mapping $T_\epsilon$ is invertible and volume-preserving, it is possible to tractably compute the density of the pushforward of a reference distribution $q(\cdot, \cdot)$ under repeated applications of it. In addition, this repeated application of $T_\epsilon$ resembles the steps of Hamiltonian Monte Carlo (HMC) [35], which we know converges in distribution to the target posterior distribution. [21, 22] use these facts to construct a normalizing flow [32] VI family. However, there are two issues with this methodology. First, the $O(Nd)$ complexity of evaluating each step $T_\epsilon$ makes training and simulating from this flow computationally expensive. Second, Hamiltonian dynamics on its own creates a flow with insufficient flexibility to match a target $\bar{\pi}$ of interest. In particular, given a density $q(\cdot, \cdot)$ and pushforward $q_t(\cdot, \cdot)$ under $H_t$, we have

$$\forall t \in \mathbb{R}, \quad \mathrm{D}_{\mathrm{KL}}\left(q_t || \bar{\pi}\right) = \mathrm{D}_{\mathrm{KL}}\left(q || \bar{\pi}\right). \tag{3}$$

In other words, Hamiltonian dynamics itself cannot reduce the KL divergence to $\bar{\pi}$; it simply interchanges potential and kinetic energy. [21] address this issue by instead deriving their flow from *tempered* Hamiltonian dynamics: for an integrable tempering function $\gamma : \mathbb{R} \to \mathbb{R}$,

$$\frac{\mathrm{d}\rho_t}{\mathrm{d}t} = \nabla \log \pi(\theta_t) - \gamma(t)\rho_t \qquad \frac{\mathrm{d}\theta_t}{\mathrm{d}t} = \rho_t. \tag{4}$$

The discretized version of the dynamics in Eq. (4) corresponds to multiplying the momentum by a tempering value $\alpha_k > 0$ after the $k^{\text{th}}$ application of $T_\epsilon$. By scaling the momentum, one provides the normalizing flow with the flexibility to change the kinetic energy at each step. However, we show later in Proposition 3.2 that just tempering the momentum does not provide the required flow flexibility, even for a simple representative Gaussian target.

A related line of work uses the mapping $T_\epsilon$ for variational annealed importance sampling [23–25]. The major difference between these methods and the normalizing flow-based methods is that the auxiliary variable is (partially) stochastically refreshed via $\rho \sim \mathcal{N}(0, I)$ after applications of $T_\epsilon$. One is then forced to minimize the KL divergence between the joint distribution of $\theta$ and all of the auxiliary momentum variables under the variational and augmented target distributions.

## 3  Sparse Hamiltonian flows

In this section we present sparse Hamiltonian flows, a new method to construct and draw samples from Bayesian coreset posterior approximations. We first present a method and supporting theory for selecting the data points to be included in the coreset, then discuss building a sparse flow with these points, and finally introduce quasi-refreshment steps to give the flow family enough flexibility to match the target distribution. Sparse Hamiltonian flows enables tractable i.i.d. sampling, provides a tractable density and normalization constant, and is constructed by minimizing the KL divergence to the posterior with simple stochastic gradient estimates.

### 3.1  Selection via subsampling

The first step in our algorithm is to choose a uniformly random subsample of $M$ points from the full dataset; these will be the data points that comprise the coreset. Without loss of generality, we assume these are the first $M$ points. The key insight in this work is that while subsampling with importance weighting does not typically provide good coreset approximations [26], a uniformly random subset of the $N$ log-likelihood potential functions $\{f_1, \ldots, f_M\}$ still provides a good *basis* for approximation with high probability. Proposition 3.1 provides the precise statement of this result for a representative example model Eq. (5). In particular, Proposition 3.1 asserts that as long as we set our coreset size $M$ to be proportional to $d \log_2 N$, the optimal coreset posterior approximation will be *exact*, i.e., have 0 KL divergence to the true posterior, with probability at least $1 - N^{-\frac{d}{2}}(\log_2 N)^{\frac{d}{2}}$. Thus we achieve an exponential compression of the dataset, $N \to \log_2 N$, without losing any fidelity. Note that we will still need a method to choose the weights $w_1, \ldots, w_M$ for the $M$ points, but the use of uniform selection rather than a one-at-a-time approach [27–29] substantially simplifies the construction. In Proposition 3.1, $C$ is the universal constant from [37, Corollary 1.2], which provides an upper bound on the number of spherical balls of some fixed radius needed to cover a $d$-dimensional unit sphere.

**Proposition 3.1.** *Consider a Bayesian Gaussian location model:*

$$\theta \sim \mathcal{N}(0, I) \quad \text{and} \quad \forall n \in [N], \quad X_n \overset{i.i.d.}{\sim} \mathcal{N}(\theta, I), \tag{5}$$

*where $\theta, X_n \in \mathbb{R}^d$ for $d \in \mathbb{N}$. Suppose the true data generating parameter $\theta = 0$, and set $M = \log_2(A_d N^d (\log N)^{-d/2}) + C$ where $A_d = e^{\frac{d}{2}} d^{\frac{3}{2}} \log(1 + d)$. Then the optimal coreset $\pi_{w^\star}$ for the model Eq.* (5) *built using a uniform subsample of data of size $M$ satisfies*

$$\limsup_{N \to \infty} \frac{\mathbb{P}\left(\mathrm{D}_{\mathrm{KL}}(\pi_{w^\star} || \pi) \neq 0\right)}{N^{-\frac{d}{2}}(\log N)^{\frac{d}{2}}} \leq 1.$$

## 3.2 Sparse flows

Upon taking a uniform subsample of $M$ data points from the full dataset, we consider the sparsified Hamiltonian dynamics initialized at $\theta_0, \rho_0 \sim q(\cdot, \cdot)$ for reference density[1] $q(\cdot, \cdot)$,

$$\frac{\mathrm{d}\rho_t}{\mathrm{d}t} = \nabla \log \pi_w(\theta_t) \qquad \frac{\mathrm{d}\theta_t}{\mathrm{d}t} = \rho_t. \tag{6}$$

Much like the original Hamiltonian dynamics for the full target density, the sparsified Hamiltonian dynamics Eq. (6) targets the augmented coreset posterior with density $\bar{\pi}_w(\theta, \rho)$ on $\mathbb{R}^{2d}$,

$$\bar{\pi}_w(\theta, \rho) \propto \pi_w(\theta) \exp\left(-\frac{1}{2}\rho^T \rho\right).$$

Discretizing these dynamics yields a leapfrog method similar to Eq. (2) with three interleaved steps,

$$\hat{\rho}_{k+1} = \rho_k + \frac{\epsilon}{2}\nabla \log \pi_w(\theta_k) \quad \theta_{k+1} = \theta_k + \epsilon\hat{\rho}_{k+1} \quad \rho_{k+1} = \hat{\rho}_{k+1} + \frac{\epsilon}{2}\nabla \log \pi_w(\theta_{k+1}). \tag{7}$$

Denote the map constructed by applying these three steps in sequence $T_{w,\epsilon} : \mathbb{R}^{2d} \to \mathbb{R}^{2d}$. Like the original leapfrog method, these transformations are both invertible and shear, and thus preserve volume; and for small enough step size $\epsilon$, they approximately maintain the invariance of $\bar{\pi}_w(\theta, \rho)$. However, since $w$ only has the first $M$ entries nonzero,

$$\nabla \log \pi_w(\theta_k) = \sum_{m=1}^{M} w_m \nabla \log f_m(\theta_k),$$

and thus a coreset leapfrog step can be taken in $O(Md)$ time, as opposed to $O(Nd)$ time in the original approach. Given that Proposition 3.1 recommends setting $M \approx d\log_2(N)$, we have achieved an exponential reduction in computational cost of running the flow.

However, as before, the weighted sparse leapfrog flow is not sufficient on its own to provide a flexible variational family. In particular, we know that $T_{w,\epsilon}$ nearly maintains the distribution $\bar{\pi}_w$ as invariant. We therefore need a way to modify the distribution of the momentum variable $\rho$. One option is to include a tempering of the form Eq. (4) into the sparse flow. However, Proposition 3.2 shows that even *optimal* tempering does not provide the flexibility to match a simple Gaussian target $\bar{\pi}$.

**Proposition 3.2.** *Let* $\theta_t, \rho_t \in \mathbb{R}$ *follow the tempered Hamiltonian dynamics Eq.* (4) *targeting* $\pi = \mathcal{N}(0, \sigma^2)$, $\sigma > 0$, *with initial distribution* $\theta_0 \sim \mathcal{N}(\mu, 1)$, $\rho_0 \sim \mathcal{N}(0, \beta^2)$ *for initial center* $\mu \in \mathbb{R}$ *and momentum scale* $\beta > 0$. *Let* $q_t$ *be the distribution of* $(\theta_t, \rho_t)$. *Then*

$$\inf_{t>0, \beta>0, \gamma:\mathbb{R}_+ \to \mathbb{R}} \mathrm{D_{KL}}\left(q_t || \bar{\pi}\right) \geq \log \frac{1 + \mu^2}{4\sigma}.$$

*Note that if* $\gamma(t) = 0$ *identically, then* $\forall t \geq 0$, $\mathrm{D_{KL}}(q_t || \bar{\pi}) = \mathrm{D_{KL}}(q_0 || \bar{\pi})$.

The intuition behind Proposition 3.2 is that while adding a tempering $\gamma(t)$ enables one to change the total energy by scaling the momentum, it does not allow one fine enough control on the distribution of the momentum. For example, if $\mathbb{E}[\rho] \neq 0$ under the current flow approximation, we cannot scale the momentum to force $\mathbb{E}[\rho] = 0$ as it should be under the augmented target; intuitively, we also need the ability to shift or recenter the momentum as well.

---

[1]The reference $q$ can also have its own variational parameters to optimize, but in this paper we leave it fixed.

| **Algorithm 1** `SparseHamFlow` | **Algorithm 2** `Estimate_ELBO` |
|---|---|
| **Require:** $\theta_0, \rho_0, w, \epsilon, \lambda, L, R$ 
 $\quad J \leftarrow 0$, and $(\theta, \rho) \leftarrow (\theta_0, \rho_0)$ 
 $\quad$ **for** $r = 1, \dots, R$ **do** 
 $\qquad$ **for** $\ell = 1, \dots, L$ **do** 
 $\qquad\quad$ Sparse flow leapfrog: 
 $\qquad\quad \theta, \rho \leftarrow T_{w,\epsilon}(\theta, \rho)$ 
 $\qquad$ **end for** 
 $\qquad$ Accumulate log Jacobian determinant: 
 $\qquad J \leftarrow J + \log\left\|\det \frac{\partial R_{\lambda_r}}{\partial \rho}(\rho, \theta)\right\|$ 
 $\qquad$ Quasi-refreshment: 
 $\qquad \rho \leftarrow R_{\lambda_r}(\rho, \theta)$ 
 $\quad$ **end for** 
 $\quad$ **return** $\theta, \rho, J$ | **Require:** $q, \pi_0, w, \epsilon, \lambda, L, R, S$ 
 $\quad (\theta_0, \rho_0) \sim q(\cdot, \cdot)$ 
 $\quad$ Forward pass: 
 $\quad \theta, \rho, J \leftarrow$ `SparseHamFlow`$(\theta_0, \rho_0, w, \epsilon, \lambda, L, R)$ 
 $\quad$ Obtain unbiased ELBO estimate: 
 $\quad (n_1, \dots, n_S) \overset{\text{i.i.d.}}{\sim} \mathsf{Unif}(\{1, 2, \dots, N\})$ 
 $\quad \log \bar{p} \leftarrow \log \pi_0(\theta) + \frac{N}{S}\sum_{s=1}^{S} f_{n_s}(\theta) +$ 
 $\qquad\qquad \log \mathcal{N}(\rho \mid 0, I)$ 
 $\quad \log \bar{q} \leftarrow q(\theta_0, \rho_0) - J$ 
 $\quad$ **return** $\log \bar{p} - \log \bar{q}$ |

### 3.3 Quasi-refreshment

Rather than resampling the momentum variable from its target marginal—which removes the ability to evaluate the density of $\theta_t, \rho_t$—in this work we introduce deterministic *quasi-refreshment* moves that enable the flow to strategically update the momentum without losing the ability to compute the density and normalization constant of $\theta_t, \rho_t$ (i.e., we construct a normalizing flow [32]). Here we introduce the notion of *marginal* quasi-refreshment, which tries to make the marginal distribution of $\rho_t$ match the corresponding marginal distribution of the augmented target $\bar{\pi}_w$. Proposition 3.3 shows that marginal quasi-refreshment is guaranteed to reduce the KL divergence.

**Proposition 3.3.** *Consider the state $\theta_t, \rho_t \in \mathbb{R}^d$ of the flow at step t, and the augmented target distribution $\theta, \rho \sim \bar{\pi}$. Suppose that we have a bijection $R : \mathbb{R}^d \to \mathbb{R}^d$ such that $R(\rho_t) \overset{d}{=} \rho$. Then*

$$\mathrm{D}_{\mathrm{KL}}\left(\theta_t, R(\rho_t)\|\theta, \rho\right) = \mathrm{D}_{\mathrm{KL}}\left(\theta_t, \rho_t\|\theta, \rho\right) - \mathrm{D}_{\mathrm{KL}}\left(\rho_t\|\rho\right).$$

See Appendix A for the proof of Proposition 3.3 and a general treatment of quasi-refreshment; for simplicity, we focus on the type of quasi-refreshment that we use in the experiments. In particular, if we are willing to make an assumption about the marginal distribution of $\rho_t$ at step $t$ of the flow, we can introduce a tunable family of functions $R_\lambda$ with parameters $\lambda$ that is flexible enough to set $R_\lambda(\rho_t) \overset{d}{=} \rho$ for some $\lambda$, and include optimization of $\lambda$ along with the coreset weights. It is important to note that this assumption on the distribution of $\rho_t$ is not related to the posterior $\pi$. As an example, in this work we assume that $\rho_t \sim \mathcal{N}(\mu, \Lambda^{-1})$ for some unknown mean $\mu$ and diagonal precision $\Lambda$, which enables us to simply set

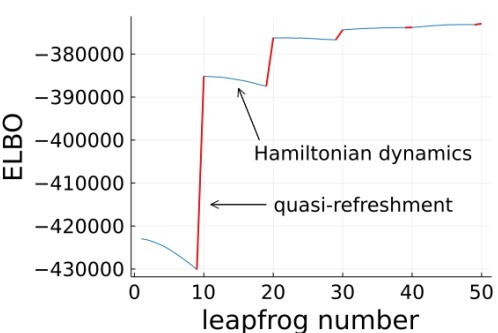

Figure 1: ELBO across leapfrog steps.

$$R_\lambda(x) = \Lambda(x - \mu). \tag{8}$$

We then include $\lambda = (\mu, \Lambda)$ as parameters to be optimized along with the coreset weights $w$ (each quasi-refreshment step will have its own set of parameters $\mu, \Lambda$). Even when this assumption does not hold exactly, the resulting form of Eq. (8) enables the refreshment step to both shift and scale (i.e., standardize) the momentum as desired, and is natural to implement as part of a single optimization routine.

Fig. 1 provides an example of the effect of quasi-refreshment in a synthetic Gaussian location model (see Section 4.1 for details). In particular, it shows the evidence lower bound (ELBO) as a function of leapfrog step number in a trained sparse Hamiltonian flow with the quasi-refreshment scheme in

Eq. (8). While the estimated ELBO values stay relatively stable across leapfrog steps in between quasi-refreshments (as expected by Eq. (3)), the quasi-refreshment steps (colored red) cause the ELBO to increase drastically. We note that the ELBO does not stay exactly constant because the Hamiltonian dynamics targets the coreset posterior instead of the true posterior, and is simulated approximately using leapfrog steps. As the series of transformations brings the approximated density closer to the target, the quasi-refreshment steps no longer change the ELBO much, signalling the convergence of the flow's approximation of the target. It is thus clear that the marginal quasi-refreshments indeed decrease the KL, as shown in Proposition 3.3.

### 3.4 Algorithm

In this section, we describe the procedure for training and generating samples from a sparse Hamiltonian flow. As a normalizing flow, a sparse Hamiltonian flow can be trained by maximizing the augmented ELBO using usual stochastic gradient methods (e.g. as in [32]), where the transformations follow Eq. (7) with a periodic quasi-refreshment. Here and in the experiments we focus on the shift-and-scale quasi-refreshment in Eq. (8).

We begin by selecting a subset of $M$ data points chosen uniformly randomly from the full data. Next we select a total number $R$ of quasi-refreshment steps, and a number $L$ of leapfrog steps between each quasi-refreshment. The flow parameters to be optimized consist of the quasi-refreshment parameters $\lambda = (\lambda_r)_{r=1}^R$, the $M$ coreset weights $w = (w_m)_{m=1}^M$, and the leapfrog step sizes $\epsilon = (\epsilon_i)_{i=1}^d$; note that we use a separate step size $\epsilon_i$ per latent variable dimension $i$ in Eq. (7) [35, Sec. 4.2]. This modification enables the flow to fit nonisotropic target distributions.

We initialize the weights to $N/M$ (i.e., a uniform coreset), and select an initial step size for all dimensions. We use a warm start to initialize the parameters $\lambda_r = (\mu_r, \Lambda_r)$ of the quasi-refreshments. Specifically, using the initial leapfrog step sizes and coreset weights, we pass a batch of samples from the reference density $q(\cdot, \cdot)$ through the flow up to the first quasi-refreshment step. We initialize $\mu_1, \Lambda_1$ to the empirical mean and diagonal precision of the samples at that point. We then apply the initialized first quasi-refreshment to the momentum, proceed with the second sequence of leapfrog steps, and repeat until we have initialized all quasi-refreshments $r = 1, \ldots, R$.

Once the parameters are initialized, we log-transform the step sizes, weights, and quasi-refreshment diagonal scaling matrices to make them unconstrained during optimization. We obtain an unbiased estimate of the augmented ELBO gradient by applying automatic differentiation [38, 39] to the ELBO estimation function Algorithm 2, and optimize all parameters jointly using a gradient-based stochastic optimization technique such as SGD [40, 41] and ADAM [42]. Once trained, we can obtain samples from the flow via Algorithm 1.

## 4 Experiments

In this section, we compare our method against other Hamiltonian-based VI methods and Bayesian coreset construction methods. Specifically, we compare the quality of posterior approximation, as well as the training and sampling times of sparse Hamiltonian flows (SHF), Hamiltonian importance sampling (HIS) [21], and unadjusted Hamiltonian annealing (UHA) [23] using real and synthetic datasets. We compare with two variants of HIS and UHA: "-Full," in which we train using in-flow minibatching as suggested by [21, 23], but compute evaluation metrics using the full-data flow; and "-Coreset," in which we base the flow on a uniformly subsampled coreset. We also include sampling times of adaptive HMC and NUTS [43, Alg. 5 and 6] using the full dataset. Finally, we compare the quality of coresets constructed by SHF to those obtained using uniform subsampling (UNI) and Hilbert coresets with orthogonal matching pursuit (Hilbert-OMP) [28, 44]. All experiments are performed on a machine with an Intel Core i7-12700H processor and 32GB memory. Code is available at https://github.com/NaitongChen/Sparse-Hamiltonian-Flows. Details of the experiments are in Appendix B.

### 4.1 Synthetic Gaussian

We first demonstrate the performance of SHF on a synthetic Gaussian-location model,

$$\theta \sim \mathcal{N}(0, I) \quad \text{and} \quad \forall n \in [N], \quad X_n \overset{\text{i.i.d.}}{\sim} \mathcal{N}(\theta, cI),$$

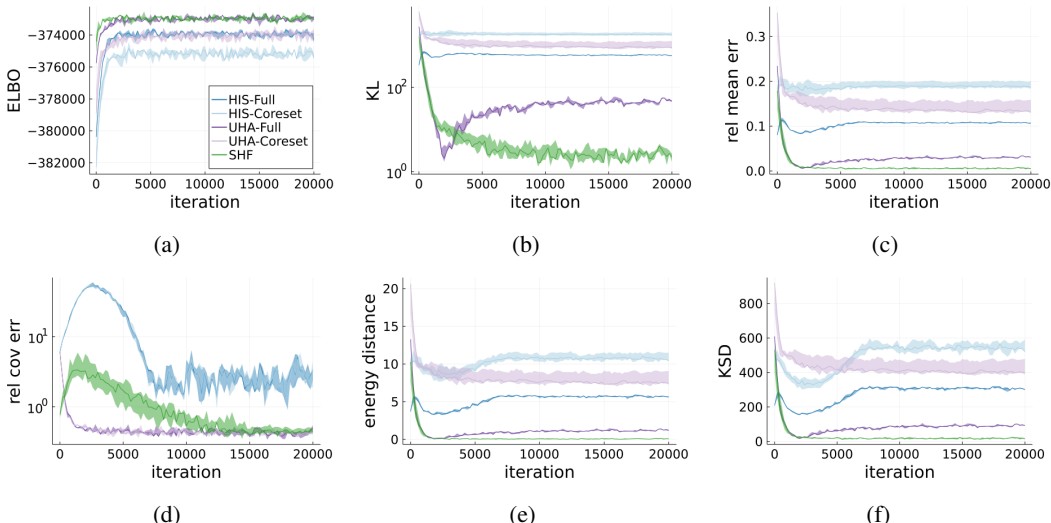

Figure 2: ELBO (Fig. 2a), KL divergence (Fig. 2b), relative 2-norm mean error (Fig. 2c), relative Frobenius norm covariance error (Fig. 2d), energy distance (Fig. 2e), and IMQ KSD [45] (Fig. 2f) for synthetic Gaussian. The lines indicate the median, and error regions indicate $25^{\text{th}}$ to $75^{\text{th}}$ percentile from 5 runs.

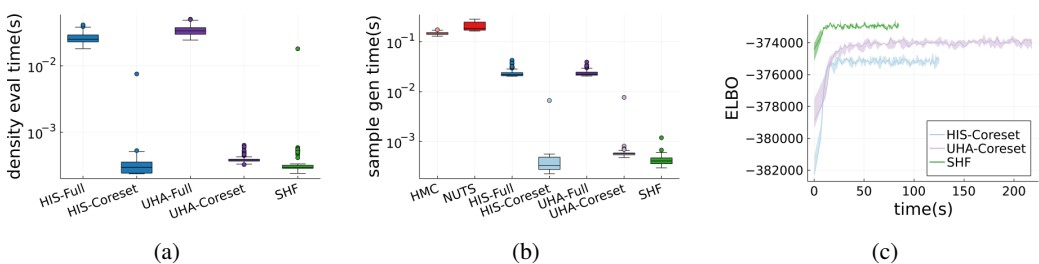

Figure 3: Density evaluation (Fig. 3a) and sample generation time (Fig. 3b) (100 samples), and ELBO versus time during training (Fig. 3c) for synthetic Gaussian. The lines indicate the median, and error regions indicate $25^{\text{th}}$ to $75^{\text{th}}$ percentile from 5 runs.

where $\theta, X_n \in \mathbb{R}^d$. We set $c = 100, d = 10, N = 10,000$. This model has a closed from posterior distribution $\pi = \mathcal{N}\left(\frac{\sum_{n=1}^N X_n}{c+N}, \frac{c}{c+N}I\right)$. More details may be found in Appendix B.1.

Fig. 2a compares the ELBO values of `SHF`, `HIS`, and `UHA` across all optimization iterations. We can see that `SHF` and `UHA-Full` result in the highest ELBO, and hence tightest bound on the log normalization constant of the target. In this problem, since we have access to the exact posterior distribution in closed form, we can also estimate the $\theta$-marginal KL divergence directly, as shown in Fig. 2b. Here we see the posterior approximation produced by `SHF` provides a significantly lower KL than the other competing methods. Figs. 2c to 2f show, through a number of other metrics, that `SHF` provides a higher quality posterior approximation than others. It is worth noting that while the relative covariance error for `SHF` takes long to converge, we observe a monotonic downward trend in both the relative mean error and KL divergence of `SHF`. This means that for this particular problem, our method finds the centre of the target before fine tuning the covariance. We also note that a number of metrics go up for `UHA-full` because it operates on the augmented space based on a sequence of distributions that bridge some simple initial distribution and the target distribution. Therefore, it is not guaranteed that all steps of optimization improve the quality of approximation on the marginal space of the latent variables of interest, which is what the plots in Fig. 2 show.

Figs. 3a and 3b show the time required for each method to evaluate the density of the joint distribution of $\theta, \rho$ and to generate samples. It is clear that the use of a coreset improves the density evaluation

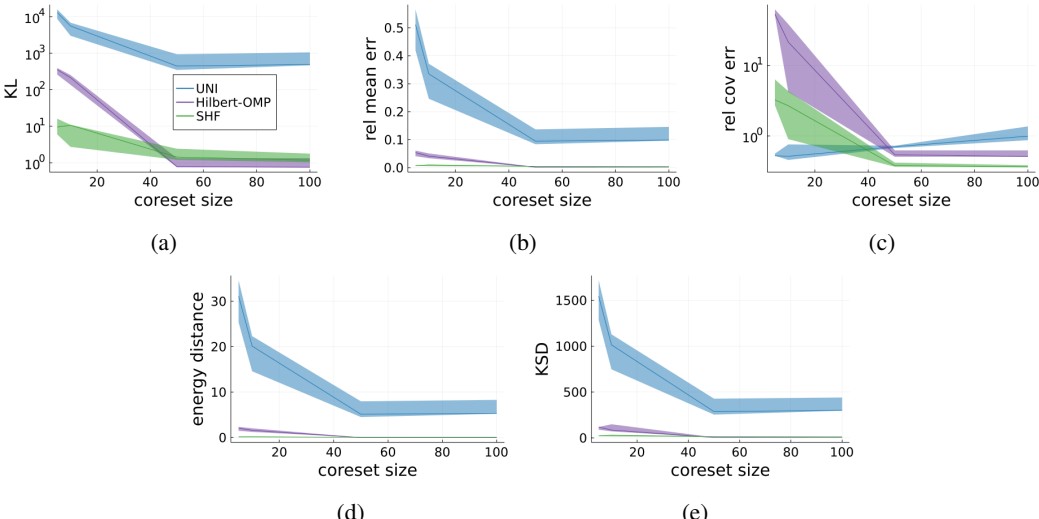

Figure 4: Estimated KL divergence (Fig. 4a), relative 2-norm mean error (Fig. 4b), relative Frobenius norm covariance error (Fig. 4c), energy distance (Fig. 4d), and IMQ KSD (Fig. 4e) versus coreset size. The lines indicate the median, and error regions indicate 25th to 75th percentile from 5 runs.

and sample generation time by more than an order of magnitude. Fig. 3c compares the training times of `SHF`, `HIS-Coreset`, and `UHA-Coreset` (recall that due to the use of subsampled minibatch flow dynamics, `HIS-Full` and `UHA-Full` share the same training time as their `-Coreset` versions). The relative training speeds generally match those of sample generation from the target posterior.

Finally, Fig. 4 compares the quality of coresets constructed via `SHF`, uniform subsampling (`UNI`), and Hilbert coresets with orthogonal matching pursuit (`Hilbert-OMP`). Note that in this problem, the Laplace approximation is exact (the true posterior is Gaussian), and hence `Hilbert-OMP` constructs a coreset using samples from the true posterior. Despite this, `SHF` provides coresets of comparable quality, in addition to enabling tractable i.i.d. sampling, density evaluation, normalization constant bounds, and straightforward construction via stochastic optimization.

## 4.2 Bayesian linear regression

In the setting of Bayesian linear regression, we are given a set of data points $(x_n, y_n)_{n=1}^N$, each consisting of features $x_n \in \mathbb{R}^p$ and response $y_n \in \mathbb{R}$, and a model of the form

$$\begin{bmatrix} \beta & \log \sigma^2 \end{bmatrix}^T \sim \mathcal{N}(0, I), \quad \forall n \in [N], \quad y_n \mid x_n, \beta, \sigma^2 \overset{\text{indep}}{\sim} \mathcal{N}\left(\begin{bmatrix} 1 & x_n^T \end{bmatrix} \beta, \sigma^2\right),$$

where $\beta \in \mathbb{R}^{p+1}$ is a vector of regression coefficients and $\sigma^2 \in \mathbb{R}_+$ is the noise variance. The dataset[2] that we use consists of $N = 100,000$ flights, each containing $p = 10$ features (e.g., distance of the flight, weather conditions, departure time, etc), and the response variable is the difference, in minutes, between the scheduled and actual departure times. More details can be found in Appendix B.2.

Since we no longer have the posterior distribution in closed form, we estimate the mean and covariance using 5000 samples from `Stan` [46] and treat them as the true posterior mean and covariance. Figs. 5a to 5c show the marginal KL, relative mean error, and relative covariance error of `SHF`, `HIS`, and `UHA`, where the marginal KL is estimated using the Gaussian approximation of the posterior with the estimated mean and covariance. Here we also include the posterior approximation obtained using the Laplace approximation as a baseline. We see that `SHF` provides the highest quality posterior approximation. Furthermore, Fig. 5d shows that `SHF` provides a significant improvement in the marginal KL compared with competing coreset constructions `UNI` and `Hilbert-OMP`. This is due to the true posterior no longer being Gaussian; the Laplace approximation required by `Hilbert-OMP` fails to capture the shape of the posterior. Additional plots comparing the quality of posterior approximations using various other metrics can be found in Appendix B.2.

---

[2]This dataset consists of airport data from `https://www.transtats.bts.gov/DL_SelectFields.asp?gnoyr_VQ=FGJ` and weather data from `https://wunderground.com`.

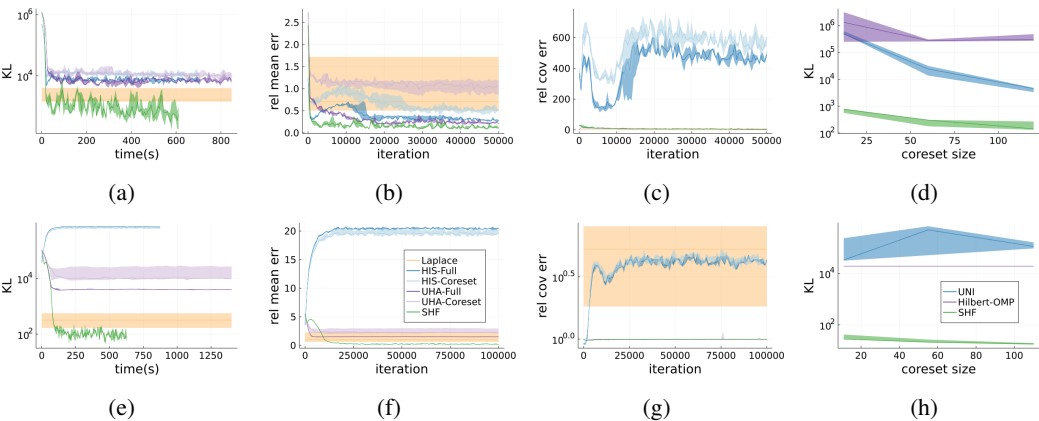

Figure 5: Linear (top) and logistic (bottom) regression results: Gaussian approximated KL divergence versus training time (Figs. 5a and 5e), relative 2-norm mean error (Figs. 5b and 5f), relative Frobenius norm covariance error (Figs. 5c and 5g), and Gaussian approximated KL divergence versus coreset size (Figs. 5d and 5h). The lines indicate the median, and error regions indicate $25^{\text{th}}$ to $75^{\text{th}}$ percentile from 5 runs.

### 4.3 Bayesian logistic regression

In the setting of Bayesian logistic regression, we are given a set of data points $(x_n, y_n)_{n=1}^{N}$, each consisting of features $x_n \in \mathbb{R}^p$ and label $y_n \in \{0, 1\}$, and a model of the form

$$\forall i \in [p+1], \ \beta_i \overset{\text{i.i.d.}}{\sim} \text{Cauchy}(0,1), \quad \forall n \in [N], \ y_n \overset{\text{indep}}{\sim} \text{Bern}\left(\left(1 + \exp\left(-\begin{bmatrix} 1 & x_n^T \end{bmatrix} \beta\right)\right)^{-1}\right),$$

where $\beta \in \mathbb{R}^{p+1}$. The same airline dataset is used with the labels indicating whether a flight is cancelled. Of the flights included, $1.384\%$ were cancelled. More details can be found in Appendix B.3.

The same procedures as in the Bayesian linear regression example are followed to generate the results in Figs. 5e to 5h. To account for the class imbalance problem present in the dataset, we construct all subsampled coresets with half the data having label $1$ and the rest with label $0$. The results in Figs. 5e to 5h are similar to those from the Bayesian linear regression example; SHF provides high quality variational approximations to the posterior. Additional plots comparing the quality of posterior approximations using various other metrics can be found in Appendix B.3.

## 5   Conclusion

This paper introduced sparse Hamiltonian flows, a novel coreset-based variational family that enables tractable i.i.d. sampling, and evalution of density and normalization constant. The method randomly subsamples a small set of data, and uses the subsample to construct a flow from sparse Hamiltonian dynamics. Novel quasi-refreshment steps provide the flow with the flexibility to match target posteriors without introducing additional auxiliary variables. Theoretical results show that, in a representative model, the method can recover the exact posterior using a subsampled dataset of the size that is a logarithm of its original size, and that quasi-refreshments are guaranteed to reduce the KL divergence to the target. Experiments demonstrate that the method provides high quality coreset posterior approximations. One main limitation of our methodology is that the data must be "compressible" in the sense that log-likelihood functions of a subset can be used to well represent the full log-likelihood. If the data do not live on some underlying low-dimensional manifold, this may not be the case. Additionally, while our quasi-refreshment is simple and works well in practice, more work is required to develop a wider variety of general-purpose quasi-refreshment moves. We leave this for future work.

## Acknowledgments and Disclosure of Funding

All authors were supported by a National Sciences and Engineering Research Council of Canada (NSERC) Discovery Grant, NSERC Discovery Launch Supplement, and a gift from Google LLC.

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
