# OpenReview forum: "Bayesian inference via sparse Hamiltonian flows"
_NeurIPS.cc/2022/Conference — NeurIPS 2022 Accept_

### Official Review · Reviewer_hoPP · 2022-06-27

**Rating:** 8
**Confidence:** 3
**Soundness:** 3 good
**Presentation:** 4 excellent
**Contribution:** 3 good

**Summary:**

“Bayesian Inference via Sparse Hamiltonian Flows” combines three techniques to make Bayesian Inference faster and more accurate. It combines a) subsampling of the data (core sets), b) sparse flows and c) quasi-refreshments.

The paper provides theoretical evidence for why these subcomponents reduce the runtime or increase performance (see section 3) and empirical evidence in three different settings. The Sparse Hamiltonian Flows (SHF) clearly and strongly outperform the alternatives in most experiments.

*UPDATE*: thanks for addressing all of my concerns. I update my score from 7 to 8. Great work. Keep it up!


**Questions:**

- Limitations: To my understanding, SHF makes some assumptions that imply limitations. For example, SHF chooses a random subset of data points. In cases where a random subset of data points is not representative of the entire dataset, SHF might be fast but not useful. Is this correct? Are there other limitations of SHF that are not explicitly mentioned?
- Scale (related to limitations): I expect that more complicated data types will run into problems with random subsets more easily. Thus SHF might not be an appropriate solution for more complex models and data. I’d like to see a more explicit interaction with these points, either by actually running an additional experiment or by stating the implicit assumptions and following consequences in more detail.
- Subsets: I’m not sure I fully understand the way in which the random subset is chosen. Is the subset of M values chosen uniformly at random once at the beginning of the process or does the subset change over time?
- Figure 1: I see that during the leapfrog steps (e.g. every 10th iteration), the ELBO jumps up towards a better state. This is in line with the theory. However, from steps 0-9 the ELBO decreases rather than increases. I don’t understand why that is the case. My naive assumption would be that the blue lines should also go up just not as drastically as the red lines.


**Limitations:**

I think, the limitations of the method were discussed insufficiently and should be addressed as described in questions 1 and 2.

I’ll use the rest of the section for high-level comments.
- In its current form, the paper convinces me that SHF decreases runtime and increases performance for datasets with low complexity. The authors show this with their theoretical analysis and empirical experiments. Furthermore, the paper is well-written and the presentation is good. All of this combined already warrants publication in my opinion.
- The assumptions that SHF makes and the implied limitations are underexplored. I expect that SHF will have a hard time with more complex models and data because it assumes that a random selection of data points is representative of the entire dataset. I think a good response or an additional experiment in this direction would convince me to raise my score further. Note, that I think the paper would be improved, even if the method is more limited than expected. Stating limitations helps readers and practitioners because it defines the scope of possible use cases more clearly.
- I want to help where I can. In case something is unclear, feel free to ask follow-up questions.


**Strengths And Weaknesses:**

In short, I think the paper is good and should be published with minor revisions.

*Strengths:*
- The paper is very well-written and clear
- The suggested combination of methods clearly and strongly improves performance compared to the alternatives
- The paper provides a theoretical analysis of why and in which manner the performance improves due to SHF.

*Weaknesses:*
- The experiments are all in fairly simple settings. The results already convince me that the method is very strong and warrants publication but a more complex experiment would increase this conviction (see questions).
- The paper says little about its limitations (see questions).

---

> ### Author Response · Authors · 2022-07-31
> **Response to Reviewer hoPP**
>
> Thank you for reviewing our manuscript, and for the positive feedback! We provide a point-to-point response to each of the comments in the review below. Don’t hesitate to follow up with any questions; we are happy to answer them.
>
> > To my understanding, SHF makes some assumptions that imply limitations. For example, SHF chooses a random subset of data points. In cases where a random subset of data points is not representative of the entire dataset, SHF might be fast but not useful. Is this correct? I expect that more complicated data types will run into problems with random subsets more easily. Thus SHF might not be an appropriate solution for more complex models and data.
>
> You are totally correct; there is usually some probability that the subsample we draw will be totally unrepresentative of the full dataset, at which point the coreset construction is flawed from the start. However, our result in Proposition 3.1 provides guidance on how large one should choose the coreset to be to avoid this problem from occurring. In particular, as long as the coreset size M is roughly d \log(N), where N is the dataset size and d is the “dimension of the log-likelihood function space,” the probability of randomly obtaining a bad subsample is quite small, as it decays at roughly a N^(-d/2) rate.
>
> Now, as you say, the data might be quite complex—in the notation of our paper, this is when that dimension “d” is quite large for the model under consideration. For example, if the data are very high-dimensional (and do not lie on a low-dimensional manifold), the value of “d” may be quite large. In these cases, we may need to use a rather large coreset, and the approach may not be so useful.
>
> We will add a discussion of this limitation (and others) in the final camera-ready version.
>
> > Is the subset of M values chosen uniformly at random once at the beginning of the process or does the subset change over time?
>
> The subset of M uniformly subsampled data points is selected once at the beginning and fixed. However, we do update the weights associated with these data points as we run the variational optimization.
>
> > from steps 0-9 the ELBO decreases rather than increases. I don’t understand why that is the case. My naive assumption would be that the blue lines should also go up just not as drastically as the red lines.
>
> Thanks for pointing this out – great observation! Actually, in theory, the ELBO should stay constant during the simulation of Hamiltonian dynamics if the simulation is perfect (see equation after line 114). However, since SHF uses the gradient of the coreset posterior rather than the full posterior to simulate the dynamics, some error will be introduced. Another source of error comes from the fact that we can only approximately simulate Hamiltonian dynamics using leapfrog steps. Both sources of error can cause the ELBO to change between quasi-refreshment steps. This error could either result in an increase (steps 30-39) or decrease (steps 0 to 9) in the ELBO.
>
> > The limitations of the method were discussed insufficiently and should be addressed.
>
> We will add some text discussing the limitations of our method for the camera-ready version. In fact, one of the limitations of our method has already been touched upon in the response to your comments above. Specifically, our method relies on the assumption that the data are compressible.

---

### Official Review · Reviewer_pCaB · 2022-07-06

**Rating:** 7
**Confidence:** 3
**Soundness:** 3 good
**Presentation:** 3 good
**Contribution:** 3 good

**Summary:**

The paper introduces a new method for constructing Bayesian coresets. The authors demonstrate that a single uniform subsampling of data points is in principle sufficient to obtain an exact coreset, and introduce the sparse Hamiltonian flow to efficiently construct and sample from the corresponding coreset posterior approximation. Notable improvements over other coreset methods are reported in several experiments.


**Questions:**

### Detailed comments and questions:

Line 110. After having introduced Hamiltonian Dynamics, the author state that here that it is possible to use it as the basis of a normalizing flow. It would be informative if they could also explain what the advantages of using a Hamiltonian flow are compared to other choices of normalizing flows. I assume this might be explained in [21,22], but it would be helpful if it was reiterated here.

Line 142. The model covered in Proposition 3.1 is referred to as a "representative example model". The authors should elaborate on why this is a representative (or even relevant) example model. Can anything be said about the compression for the posterior of other models? Can anything be said for finite N? In particular, it would be interesting to know whether anything could be said about the choice of M in the general setting.

Line 197. "In this work we assume that p_t ~ N(mu, Lambda^-1)"
Does this mean that we are ultimately assuming that the posterior can be approximated by a diagonal Gaussian? In that case, how is this different from the assumption made in a simple parametric VI setting? In the introduction, the authors explicitly mention the “simple parametric families” as a contrast to the current work. It would be helpful if the contrast between the two was explicitly explained for the case when this parametric assumption of p_t is made.

Line 266. "Figs. 2c and 2d demonstrate that this reduction in KL divergence is primarily due to a lower relative error in the approximate posterior mean provided by SHF."
To which extent is this explained by the fact that the posterior is Gaussian in this case? It seems like an idealized setting for the proposed method, given the assumption of Gaussianity of p_t.

In the Experiments section, the choice of baselines used in the different figures was not clear to me. Why there was a difference in the baseline methods included in Fig 2/3 and Fig 4?

I would have liked to see simpler Bayesian Inference baselines such as Laplace and simple mean-field VI included as baselines as well. This would make it easier for someone not intimately familiar with coreset methods to judge how big an impact these methods have over conventional approaches.

In the Conclusion section, it would have been helpful with a discussion of the limitations of the proposed approach.


### Minor details

Line 81. "While theoretically not expensive, interleaving MCMC and gradient descent steps is hard to implement and tune, and is too slow to be practical. "
This sentence was not entirely clear to me. First, you state that it is theoretically not expensive, and that it is hard to implement and tune, but then you conclude that it is too slow to be practical, which seems to contradict the first part of the sentence. Please clarify whether the limitation here is fundamental (e.g. efficiency wise), or whether it is practical (difficult to implement and tune).

Line 149. "Let C be the universal constant from [37], Corollary 1.2.
This constant appears a bit unmotivated here right before Propoposition 3.1. It might increase readibility if it was either explained in greater detail, or otherwise moved to the end of the sentence in 152.

Line 163. "since w only has the first M entries nonzero"
Was the convention of placing the nonzero elements first mentioned anywhere? Perhaps I missed it.


**Limitations:**

The authors do not discuss the limitations of their method.

**Strengths And Weaknesses:**

### Originality:
To my knowledge, the presented method is original. The authors provide a footnote citing concurrent work based on similar ideas, but the combination of Hamiltonian flow approximations to the coreset posterior seems unique to his work. One could argue that the authors should provide a dedicated Related Work section to elaborate on the connections to earlier work.

### Quality:
The paper is technically sound, and the claims are carefully developed and well supported. The paper could be further improved with some reflection on the limitations of the approach.

### Clarity:
The manuscript is well structured and very clearly written, with helpful introductions to the methodological ingredients that it builds upon.

### Significance:
The paper constitutes a significant contribution within research on Bayesian coresets, both in terms of methodology and measured in terms of the performance improvements over other methods. I am not certain how large a contribution it will have to the field of Bayesian inference in general. This would have been easier to assess if the authors had broadened the scope of their baselines to other Bayesian inference procedures.

---

> ### Author Response · Authors · 2022-07-31
> **Response to Reviewer pCaB**
>
> Thank you for your efforts in reviewing our manuscript, and for the positive feedback! We provide a point-to-point response to each of the comments in the review below. Don’t hesitate to follow up with any questions; we are happy to answer them.
>
> > ...the authors should provide a dedicated Related Work section to elaborate on the connections to earlier work.
>
> You are correct that the combination of Hamiltonian-based flow with quasi-refreshment and Bayesian coresets is unique to our work. We can certainly add a dedicated related work section that draws connections to earlier work for the camera-ready version.
>
> > ...explain what the advantages of using a Hamiltonian flow are compared to other choices of normalizing flows.
>
> This is a great suggestion! The intuition here is that Hamiltonian flow methods involve a sequence of transformations that resemble the steps of Hamiltonian Monte Carlo (HMC). In HMC, we run Hamiltonian dynamics for a while, and then resample the momentum, and repeat; in Hamiltonian flows, we run Hamiltonian dynamics for a while, and then quasi-refresh the momentum, and then repeat. Since we know the Markov chain generated by HMC converges (in distribution) to the target posterior distribution, we expect the Hamiltonian flow to do something similar (although not perfectly, as the quasi-refreshment is not a perfect substitute for exact momentum resampling!).
>
> One other very important advantage of Hamiltonian flows is that the ODE naturally enables the use of a coreset, which enables us to build a computationally inexpensive flow. By the previous argument, our coreset flow should approximate the coreset posterior reasonably well. Moreover, the optimal coreset should provide a good approximation of the full posterior (for the Gaussian location model this is given by our Proposition 3.1). Therefore, we have constructed a variational family that is both inexpensive to work with and flexible enough to well-approximate the posterior. Standard normalizing flows typically come with no guarantees, and may or may not be expensive to work with depending on design (though more flexible families tend to be more expensive).
>
> We will add some text outlining these advantages for the camera-ready version.
>
> > ...why this is a representative (or even relevant) example model. Can anything be said about the compression for the posterior of other models? Can anything be said for finite N?
>
> Great question! By “representative,” we mean in terms of how well our result regarding the optimal coreset quality extends. In fact, our proof technique has already been extended to general exponential families (the only difference lies in the choice of sufficient statistic). Unfortunately, this result is in a forthcoming unpublished manuscript which we can’t provide a pointer to quite yet! But if you are willing to take that leap of faith, we note that exponential families can be used to well-approximate a very wide range of models. So we suspect that our result is indicative of a more general setting (at least the fact that the coreset size should scale with some notion of a “dimension” of the model).
>
> Regarding finite N, our proof actually provides a finite-N analysis. See the proof in Appendix C (line 573). However, the finite-N result is somewhat complicated and messy. We chose to present the asymptotic version of the result in the main text due to the simpler final expression.
>
>
> > "In this work we assume that \rho_t ~ N(mu, Lambda^-1)" Does this mean that we are ultimately assuming that the posterior can be approximated by a diagonal Gaussian?
>
> No, we do not assume the posterior is a diagonal Gaussian. Let us clarify a bit with two points.
>
> 1. The posterior \pi is set by the likelihood and prior, and is usually not Gaussian. We then augment the posterior with a Gaussian distribution for the momentum. Note that this is not limiting; we are free to pick any "nice" distribution for the momentum component. The Gaussian distribution is what people typically use in practice for HMC, and we follow that standard here.
>
> 2. The variable \rho_t is the momentum variable at time t of the Hamiltonian flow. So when we say we assume \rho_t ~ N(\mu, \Lambda^-1), we are making an assumption on the momentum component at time t of our flow.
>
> The assumption is true for all t if the posterior \pi is indeed Gaussian. But we actually don’t need this assumption in practical application. The method still works without it, and applies to a wide range of non-Gaussian posteriors. The reason we included this text is because if the assumption holds, the “standardization” quasi-refreshment in equation (7) is guaranteed to reduce the KL by Proposition 3.3.  This assumption usually will not hold exactly in practice, but we have seen that the standardization quasi-refreshment still provides a significant reduction in KL (Figure 1).
>
> We will revise the wording in this section for the camera-ready version.

---

> > ### Author Response · Authors · 2022-07-31
> > **Continued Response to pCaB**
> >
> > > "Figs. 2c and 2d demonstrate..." To which extent is this explained by the fact that the posterior is Gaussian in this case?
> >
> > The quote is just a small remark regarding the plots for this particular example; we did not intend to suggest drawing conclusions outside of this Gaussian location model setting. But we recognize that this may have been implied, and will carefully re-word the text for the camera-ready. Finally, note that we see from Fig 5 that our method outperforms other competing methods under models with non-Gaussian posteriors.
> >
> > > Why was there a difference in the baseline methods included in Fig 2/3 and Fig 4?
> >
> > This work builds upon two lines of literature: coreset construction methods, and Hamiltonian-based VI methods. Each of these plots shows a comparison with methods from each line of past work. Figs. 2-3 compare the sample quality, density evaluation time and sample generation time across the posterior approximations produced by various Hamiltonian-based VI methods. Fig. 4 then compares the quality of the posteriors approximated using the coreset obtained from SHF against other coreset construction methods.
> >
> > For Figure 2-3 specifically, it is worth noting that all previous Hamiltonian VI methods do not work with a Bayesian coreset; hence they are often quite slow in the regime of large-scale data. In contrast, SHF incorporates coreset Hamiltonian dynamics, leading to fast training, density evaluation, and i.i.d. sampling.
> >
> > For Figure 4 specifically, it is worth noting that all previous coreset construction methods do not enable i.i.d. sampling: one needs to use a subsequent inference method on the coreset posterior after construction. In contrast, SHF does enable i.i.d. sampling and density evaluation with no additional secondary stage.
> >
> > > I would have liked to see simpler Bayesian Inference baselines such as Laplace and simple mean-field VI included as baselines as well.
> >
> > Thank you for the suggestion. We have added a comparison between our method and Laplace approximation using the same Bayesian linear regression model (Section 4.2) in the supplement in Appendix F. Fig. 8 shows that our method provides a higher quality posterior approximation than the Laplace approximation. Specifically, the approximated KL is around 100 for our method, and around 500 for the Laplace approximation. In general, since Laplace and mean-field VI both use Gaussian distributions to approximate the posterior, we anticipate that these simpler baselines will suffer when the posterior is non-Gaussian.
> >
> >
> > > The paper could be further improved with some reflection on the limitations of the approach.
> >
> > We agree that there was not enough attention devoted to this in the current version, and will include a discussion in the camera-ready version. One main limitation of this methodology is that we assume that the data are "compressible" in the sense that log-likelihood functions of a subset can be used to represent the full log-likelihood (see our response to comments by Reviewer hoPP). If the data are truly very high-dimensional (i.e. not on some low-dimensional manifold), this may not be the case. Another limitation is that while our quasi-refreshment is simple and works well in practice, more work is required to develop quasi-refreshment methods with general guarantees.
> >
> > > "...interleaving MCMC and gradient descent steps... " This sentence was not entirely clear to me.
> >
> > We agree that this could be stated more clearly. The idea is: at each iteration of the optimization, MCMC samples for the current coreset posterior are used to estimate the coreset weight gradient. Since the coreset size (M) is much smaller than that of the full dataset (N), MCMC is not expensive. However, the quality of the MCMC samples may be poor without tuning. And it is not realistic to tune the MCMC sampler at each iteration. We will reword this sentence to make it clearer.
> >
> > > "Let C be the universal constant" ...appears a bit unmotivated here
> >
> > The constant C here provides an upper bound on the number of spherical balls needed to cover a d-dimensional unit sphere. It is a technical detail that is not of significant importance. We will revise the paper to provide some more context on this constant.
> >
> > > "since w only has the first M entries nonzero" Was the convention of placing the nonzero elements first mentioned anywhere?
> >
> > We mentioned this in line 141; this assumption is merely for notational convenience. We will find a way to make it more prominent in the text!

---

> > > ### Comment · Reviewer_pCaB · 2022-08-09
> > > **Thanks for the clarifications**
> > >
> > > Thanks for the detailed response to my questions. It has improved my understanding of the paper. I look forward to reading the camera-ready version, containing the edits you describe. I will be changing my score to a 7.
> > >
> > > If you can find the time, I have one remaining clarifying question (I realize that we are close to the deadline):
> > >
> > > In your explanation of the advantages of using a Hamiltonian Flow over other methods, you highlight 1) the ability to converge to a target posterior distribution, and 2) that it enables the use of a core set. I'm probably missing something here, but aren't these two properties valid for any choice of normalizing flow? Do I understand correctly that the main advantage is the fact that you 1) can provide guarantees in the form of prop 3.1, and 2) that it is computationally efficient? So, in principle, it might be the case that an different choice of (expressive) flow might provide better results empirically, possibly at a higher computational cost?

---

> > > > ### Author Response · Authors · 2022-08-09
> > > > **Continued Response to pCaB**
> > > >
> > > > Thanks for the follow-up!
> > > >
> > > > Your understanding of one of the main strengths of our work is correct: the use of a coreset enables our method to be computationally efficient, and Prop 3.1 shows how big the coreset must be to enable an accurate reproduction of the full posterior.
> > > >
> > > > Generic normalizing flows (Sylvester, planar, etc.) actually cannot use coresets. Generic flows are constructed of black-box parametrized transformations---the flow structure itself does not use information from the target. In contrast, our sparse Hamiltonian flow directly incorporates the coreset target information (see line 156, eq. 5). This allows us to train the coreset weights.
> > > >
> > > > Generic normalizing flows also do not generally provide error guarantees (in the sense of "longer flow provides a lower KL"). There is some past work on analyzing how expressive these families are, but these results are usually abstract universal approximation results (see, e.g., Theorem 3.1, "The Expressive Power of a Class of Normalizing Flow Models", 2020).
> > > >
> > > > It is certainly possible for a generic normalizing flow to outperform our method on a given problem; but without guarantees, it's hard to say much in advance! (We have also found in our experience that it is hard to train generic flows reliably in practice; they often get stuck in bad local optima.)

---

### Official Review · Reviewer_ofer · 2022-07-10

**Rating:** 9
**Confidence:** 3
**Soundness:** 4 excellent
**Presentation:** 4 excellent
**Contribution:** 4 excellent

**Summary:**

This paper proposes a Bayesian inference methodology incorporating coresets with Hamiltonian flows. The paper demonstrates theoretically the challenges that both coresets and variational inference via Hamiltonian dynamics face, and proposes a fix for both in their algorithm "Sparse Hamiltonian Flows". Their method first selects a coreset and then follows a sparsified Hamiltonian flow with quasi-refreshments, which allows the flow to update the momentum. Important or argumentative claims are backed up with theoretical proofs. Experiments on a variety of regression problems demonstrate the superiority of their algorithm over current state-of-the-art coreset compression and variational-flow-based methods.

**Questions:**

- I am under the impression that the quasi-refreshment step could also be incorporated into some of the other baselines considered in the experiments, and not just for SHF. If this is the case, how would the baselines then perform?
- In Figure 2, relative cov error seems to take a long time to converge, compared to UHA-Full. Why did the purple UHA-Full suddenly go up after ~2000 iterations? It seems like at that point, the approximation is just as good as SHF.

**Limitations:**

Yes

**Strengths And Weaknesses:**

Strengths:
- Important claims and new insights on coresets and Hamiltonian flows are backed up with theoretical proofs. Where may be difficult to prove in the general case, such as Proposition 3.1, a representative example is given and the claim is proven on it.
- The proposed idea is very novel and addresses important drawbacks that current coreset methods suffer from.
- A thorough and clear review of related and past methods is provided
- Experiments are well-conducted and a variety of datasets, both synthetic and real, are explored

Weaknesses:
- Perhaps some other choices of $\rho_t$ and $R_\lambda(x)$ could be explored so that a potential user could understand the sensitivity of the algorithm to these choices

---

> ### Author Response · Authors · 2022-07-31
> **Response to Reviewer ofer**
>
> Thank you for reviewing our manuscript, and for your encouraging feedback! We provide a point-to-point response to each of the comments in the review below. Don’t hesitate to follow up with any questions; we are happy to answer them.
>
> > Perhaps some other choices of \rho_t and R_\lambda(x) could be explored so that a potential user could understand the sensitivity of the algorithm to these choices
>
> Thank you for your suggestion. Indeed, there are many possible choices of the momentum distribution for Hamiltonian dynamics. We chose the Gaussian momentum in the paper because it is the most commonly used in practice, and it enables a simple quasi-refreshment scheme of standardizing the momentum variables. We would like to point out that a discussion on some other possible quasi-refreshment schemes for Gaussian momentum was included in the supplement in Appendix A. In our experiments, we did not observe major differences in performance across the various schemes discussed, and so opted for the simplest one.
>
> As our work provides a general framework for incorporating momentum refreshments in Hamiltonian-based flow methods, however, it would likely not be too onerous to try out previously-studied alternative Hamiltonian momentum distributions, e.g. the Laplace distribution [1-2]. Developing quasi-refreshment schemes for other momentum distributions is certainly an interesting direction to explore; we leave this to future work.
>
> > I am under the impression that the quasi-refreshment step could also be incorporated into some of the other baselines considered in the experiments, and not just for SHF. If this is the case, how would the baselines then perform?
>
> You are absolutely correct that we can incorporate the quasi-refreshment step into HIS. In fact, one way to think of our new method is that we (1) replace the tempering step in HIS with quasi-refreshment, and (2) introduce the use of a coreset. Indeed, the quasi-refreshment step is precisely motivated by Proposition 3.2, which states that tempering alone is insufficient for HIS to obtain adequate target approximations, even when they are Gaussian. For UHA specifically, there is no need to incorporate the quasi-refreshment step; the momentum variables are already perfectly refreshed, as they are resampled from a Gaussian (at the cost of introducing many auxiliary variables).
>
> > In Figure 2, relative cov error seems to take a long time to converge, compared to UHA-Full. Why did the purple UHA-Full suddenly go up after ~2000 iterations? It seems like at that point, the approximation is just as good as SHF.
>
> Indeed the relative covariance error seems slow to converge, but this is not the complete picture. We need to look at the relative covariance error plot together with the relative mean error plot (Fig. 2c) and the KL plot (Fig. 2b). In particular, the relative mean and covariance error plots depict two different aspects of the quality of our target approximation; the KL plot takes both of these into consideration. For this particular problem, our method finds the center of the target before fine tuning the covariance. The monotonic downward trend of the KL divergence shows that our method keeps on improving the target approximation throughout optimization.
>
> To understand why the relative mean error and KL divergence go up for UHA, it is important to note that UHA operates on the augmented space based on a sequence of distributions that bridge some simple initial distribution and the target distribution. Therefore, it is not guaranteed that all steps of optimization improve the quality of approximation on the marginal space of the latent variables of interest. This explains the increase in the error metrics on the \theta-marginal space shown in Figs. 2b and 2c. However, we note that from Fig 2a, UHA’s augmented ELBO as the optimization objective that we maximize over shows a monotonic increasing trend.
>
> We realize that we have not been very clear about these interpretations in the paper, and will add some clarification on this in the experiment section for the camera-ready version. Thank you for pointing this out!
>
> [1] Zhang, Y. et al (2016). Laplacian Hamiltonian Monte Carlo. In: Machine Learning and Knowledge Discovery in Databases. ECML PKDD 2016. Lecture Notes in Computer Science(), vol 9851.
>
> [2] Nishimura, A. et al. Discontinuous Hamiltonian Monte Carlo for discrete parameters and discontinuous likelihoods, Biometrika 107(2), 2020, Pages 365–380.

---

> > ### Comment · Reviewer_ofer · 2022-08-08
> > **Thank you for the authors' response**
> >
> > Many thanks for your response and clarification on the points raised! I just have one more question:
> >
> > > Indeed the relative covariance error seems slow to converge, but this is not the complete picture. We need to look at the relative covariance error plot together with the relative mean error plot (Fig. 2c) and the KL plot (Fig. 2b). In particular, the relative mean and covariance error plots depict two different aspects of the quality of our target approximation; the KL plot takes both of these into consideration. For this particular problem, our method finds the center of the target before fine tuning the covariance. The monotonic downward trend of the KL divergence shows that our method keeps on improving the target approximation throughout optimization.
> >
> > > To understand why the relative mean error and KL divergence go up for UHA, it is important to note that UHA operates on the augmented space based on a sequence of distributions that bridge some simple initial distribution and the target distribution. Therefore, it is not guaranteed that all steps of optimization improve the quality of approximation on the marginal space of the latent variables of interest. This explains the increase in the error metrics on the \theta-marginal space shown in Figs. 2b and 2c. However, we note that from Fig 2a, UHA’s augmented ELBO as the optimization objective that we maximize over shows a monotonic increasing trend.
> >
> > Regarding this, in addition to looking at the relative mean and cov separately, perhaps it would be clearer if you plotted the energy distance or MMD?

---

> > > ### Author Response · Authors · 2022-08-08
> > > **Response to Reviewer ofer**
> > >
> > > Thank you again for your suggestion! We will include the comparisons in both energy distance and MMD in the supplement for the camera-ready version.

---

### Meta-Review · Area_Chair_4R4G · 2022-08-24

**Recommendation:** Accept
**Confidence:** Certain

**Metareview:**

All reviewers agree that the paper proposes an interesting approach to Bayesian inference incorporating coresets with Hamiltonian flows. Although some reviewers have some technical concerns at their first reviews, basically those have been resolved by the authors' responses. Thus, although there are some points that should be modified from the current form, I think we can expect the authors modify the paper in the camera-ready by reflecting the discussion. Based on these, I recommend acceptance for this paper.

**Award:**

No

---

### Decision · Program_Chairs · 2022-09-14

Accept